# Omega-3 Supplementation and Nutritional Status in Patients with Pancreatic Neoplasms: A Systematic Review

**DOI:** 10.3390/nu16234036

**Published:** 2024-11-26

**Authors:** Luciana Bicalho Cevolani Pires, Luciane Bresciani Salaroli, Olívia Perim Galvão de Podesta, Fabiano Kenji Haraguchi, Luís Carlos Lopes-Júnior

**Affiliations:** Graduate Program in Nutrition and Health, Health Sciences Center, Federal University of Espírito Santo (UFES), Vitória 29047-105, ES, Brazil

**Keywords:** omega-3 fatty acids, pancreatic neoplasms, nutritional status, systematic review

## Abstract

Objectives: The purpose of this study was to synthesize and evaluate the evidence regarding the effects of omega-3 supplementation on the nutritional status of pancreatic cancer patients. Methods: A systematic review of clinical trials was conducted, adhering to the PRISMA Statement. MEDLINE/PubMed, EMBASE, CENTRAL Cochrane, Scopus, and Web of Science databases were searched up to 31 December 2022 without restrictions on the publication date or language. Independent reviewers extracted data and assessed the risk of bias. The internal validity and risk of bias in randomized controlled trials (RCT) were assessed using the revised Cochrane risk of bias tool for randomized trials-RoB2, while the risk of bias in non-randomized intervention studies was evaluated using the ROBINS-I tool. Results: Eight studies met all the inclusion criteria and were analyzed. Five of them were RCT, with the majority (*n* = 4) classified as low risk of bias, and the three quasi-experiments were deemed to have a moderate risk of bias. Among the studies investigating the outcome of weight gain/maintenance, six reported statistically significant positive results (*p* < 0.05). Conclusions: In conclusion, the presented evidence indicates that omega-3 supplementation in pancreatic cancer patients is safe, well-tolerated, and beneficial, as it contributes to the stabilization or increase in body weight, as well as a reduction in inflammatory biomarkers.

## 1. Introduction

Pancreatic cancer accounts for nearly 2.6% of all new cancer cases among both men and women [1,2]. Nonetheless, it is the seventh leading cause of cancer-related deaths in both sexes, representing 4.7% of all cancer deaths, which amounts to more than 466,003 deaths worldwide [1].

Among the risk factors for pancreatic cancer, notable ones include diabetes [3], high body mass index (BMI) [4], excessive alcohol consumption [5], and tobacco use [6]. Other risk factors correlating with pancreatic cancer incidence include a family history of cancer [7], pancreatitis [8], and genetic mutations, particularly in the BRCA1 and BRCA2 genes [9,10,11,12].

Patients with pancreatic cancer typically experience weight loss of over 10% of their body weight within six months [13]. As a result, the majority of these patients develop malnutrition [14], which can progress to cachexia [15]. The severity of these consequences is a significant risk factor for disease progression and overall low survival [16,17,18].

The weight loss associated with cancer cachexia is primarily due to a decrease in muscle mass, which may or may not be accompanied by loss of fat mass, affecting over half of oncology patients [16,18,19]. However, it cannot be fully reversed by conventional nutritional therapy, leading to progressive deterioration of the body composition [20,21].

According to the guidelines established by the European Society for Clinical Nutrition and Metabolism (ESPEN), with the publication of clinical nutrition guidelines in cancer, a comprehensive assessment of nutritional status should be conducted soon after a patient undergoes oncological care [22]. Continuous monitoring is recommended, followed by the implementation of nutritional interventions regardless of the stage of the oncological disease [22,23].

For this assessment of nutritional status, the following methods are suggested by the European Society for Medical Oncology (ESMO): body composition, BMI, dietary intake, C-reactive protein, albumin, systemic inflammation, subjective global assessment (SGA), and patient-generated subjective global assessment (PG-SGA) [24].

In cancer, tumor cells synthesize pro-inflammatory cytokines and acute-phase proteins that modulate cancer cachexia by triggering proteolysis, leading to skeletal muscle depletion, and lipolysis, resulting in loss of adipose tissue mass [25]. Omega-3 fatty acids supplementation, including eicosapentaenoic acid (EPA) and docosahexaenoic acid (DHA), has been utilized as a preventive and therapeutic strategy for cachexia by reducing the production of inflammatory eicosanoids, cytokines such as IL-6 and IL-8, and NFκB activation [26]. This supplementation has been shown to increase food intake, promote weight gain, and improve body composition in patients [19,23,27].

Omega-3 fatty acids are polyunsaturated fatty acids (PUFAs) characterized by the presence of a double bond three atoms away from the terminal methyl group in their chemical structure [26]. They are widely found in natural oils of edible seeds and marine products and play an important role in lipid metabolism in humans. Omega-3 fatty acids are shown to suppress systemic inflammatory and oxidative responses, improve the appetite of patients, and enhance weight gain in cachectic patients with cancer [26,27].

Although systematic reviews have investigated the supplementation of omega-3 fatty acids and the improvement of nutritional status in various types of cancer, including head and neck cancer [28], gastrointestinal malignancy surgery [29], prostate cancer [30], and lung cancer undergoing radiotherapy and chemotherapy [31], there is still a lack of summarized evidence regarding the role of omega-3 in the nutritional status of pancreatic cancer patients.

It is noteworthy that because the pancreas is an organ of the gastrointestinal tract that produces digestive enzymes [32]; therefore, as a result, patients with pancreatic cancer will have a compromised nutrient metabolism, which leads to considerable weight loss with significant influence on nutritional status [33]. In addition, the treatment of these patients may place metabolic demands that will exacerbate any existing nutritional imbalances [34]. Therefore, here, we aimed to synthesize and evaluate the scientific evidence with regard to the effects of omega-3 supplementation on the nutritional status of pancreatic cancer patients.

## 2. Methods

### 2.1. Study Design

This is a systematic review of intervention studies [35] that adheres closely to the Preferred Reporting Items for Systematic Reviews and Meta-Analyses (PRISMA) Statement 2020. The review was conducted to investigate the effect of omega-3 supplementation on the nutritional status of patients with pancreatic cancer. To ensure transparency in the conduct of this systematic review, and following the recommendation of the Cochrane Collaboration [35], the protocol was registered with the International Prospective Register of Systematic Reviews (PROSPERO/UK) (Registration ID: CRD42022332619), and subsequently, the study protocol was published elsewhere [36].

### 2.2. Search Strategy

The search was conducted in five electronic databases MEDLINE/PubMed, Cochrane Central Register of Controlled Trials (CENTRAL) Cochrane, EMBASE, Web of Science, and SCOPUS as described in detail on study protocol [36]. The searches were updated on 31 December 2022. The search strategy for identifying studies consisted of a combination of controlled descriptors (such as MeSH terms, Emtree terms), synonyms, and keywords, as indicated in each database [37]. Boolean operators “AND”, “OR”, and “NOT” were used to combine the descriptors [38,39]. No language or time restrictions were applied. In addition to the aforementioned electronic databases, we searched clinical trial registries such as ClinicalTrials.gov, the WHO ICTRP, as well as conducted additional searches on organization websites and websites such as The British Library, Google Scholar, and preprints from medRxiv. Furthermore, we examined the reference lists of previously selected studies to identify additional relevant papers [40].

During the search strategy phase, the EndNoteweb was utilized to store, organize, and manage all the retrieved studies. The complete search strategy for each database and additional sources are detailed in Appendix A.

### 2.3. Eligibility

The selected studies followed the inclusion criteria based on the PICOS framework [41] as follows: primary studies derived from RCT and/or NRCT (quasi-experiments), assessing the effect of omega-3 supplementation in adult patients (>18 years) of both sexes and any ethnicity diagnosed with pancreatic cancer with the primary outcome being the nutritional status of the patient. Experimental studies conducted on animal models, in vivo and ex vivo studies on the topic were excluded, as well as observational studies, qualitative studies, reviews, and gray literature.

The studies were independently and blindly selected by 2 reviewers (LBCP and LCLJ). A third reviewer analyzed and made decisions regarding the inclusion or exclusion of articles with conflicting decisions [42]. During this stage of article inclusion and exclusion, the Rayyan™app (https://www.rayyan.ai/about-us/, accessed on 30 October 2024) [43] was used as a tool to assist in the eligibility phase.

### 2.4. Data Extraction

Initially, the screening of studies was based on their titles and abstracts and was performed independently by 2 researchers (LBCP and LCLJ), using previously published data extraction forms [37,39,44,45,46,47,48,49,50,51], as described in detail on study protocol [36].

### 2.5. Methodological Assessment of the Studies

The evidence level was classified using the Oxford Centre for Evidence-Based Medicine [52]. The internal validity and risk of bias of RCT were assessed using the revised RoB 2 [53]. This tool evaluates the risk of bias across five domains: (1) randomization process; (2) deviations from intended interventions; (3) missing outcome data; (4) outcome measurement; and (5) selective outcome reporting [53]. For the assessment of quasi-experimental studies, the ROBINS-I tool was used [54]. ROBINS-I comprises seven domains of bias organized chronologically in pre-intervention, during intervention, and post-intervention [54].

### 2.6. Data Synthesis

Due to the heterogeneity in methods, dosages, and protocols of omega-3 supplementation, we employed a descriptive vote counting approach along with a narrative synthesis to summarize the results [55,56]. For the narrative synthesis, we summarized the articles within each primary outcome category according to two main areas, namely omega-3 supplementation and nutritional status.

## 3. Results

### 3.1. Study Selection

The literature search across the five selected databases, as well as the additional searches in other sources, identified a total of 164 studies. Among these, 83 studies were identified as duplicates in the EndNote™ reference manager and were subsequently removed. A total of 81 studies proceeded to the title and abstract screening phase. In this stage, conducted using the Rayyan™app, 55 articles were excluded as they did not meet the inclusion criteria. The exclusion based on the title and abstract resulted in the pre-selection of 26 studies, which were then subjected to thorough full-text reading. Following this step, 21 studies were excluded as they did not fully meet the inclusion criteria and therefore did not address the research question directly [Appendix A]. Additional searches were also conducted, identifying 19 articles, of which 16 were excluded for not meeting the inclusion criteria. Hence, a total of 8 studies were included for qualitative synthesis and analysis, as depicted in Figure 1.

### 3.2. Characterization of the Studies

The eight selected articles provided data on a total of 588 patients, ranging in age from 50 to 78 years. Out of the eight studies, five were RCTs [57,58,59,60,61] and three were quasi-experimental studies [62,63,64]. The publication dates of the studies ranged from 1999 to 2019, and all were published in the English language. These studies were conducted in four different countries: Japan, Germany, Australia, and Scotland (Table 1). 

### 3.3. Omega-3 Supplementation and Nutritional Status

With regard to the nutritional status of patients with pancreatic cancer supplemented with omega-3, the most commonly used outcomes for assessing the nutritional status shown in this systematic review includes: weight, body composition, serum albumin, prealbumin, transferrin, ratio EPA/AA, skeletal muscle mass, fat percentage, interleukin-6, BMI, psoas major muscle area (PMA), lymphocytes, total cholesterol, triglycerides, c-reactive protein, HbA1c (%), cortisol, insulin, skinfold arm, back, and iliac crest.

As for the form of omega-3 supplementation evaluated, out of the eight included studies, two [59,64] used capsules, while the other six [57,58,60,61,62,63] involved the consumption of liquid nutritional supplements. The amount of EPA varied between 300 mg [59] and 6 g [64] per day during the nutritional intervention.

It is worth noting that the study by Akita et al. [58] was the only one that, in addition to EPA supplementation, included three nutritional consultations. The study also reported that many patients (54.83%) were unable to consume the recommended amount of the EPA-enriched supplement due to its taste [58]. On the other hand, in the study by Ashida et al. [57], the supplement used contained 2 g of EPA and was the only one among the eight studies that, in addition to the supplement, provided 1200 kcal of regular food intake for seven days before the operation.

A prospective study with 200 patients undergoing treatment for gastrointestinal and hepatobiliary cancer, which investigated the underdiagnosis of malnutrition, endorses the need for referral to an oncology nutritionist—a pivotal health professional in the treatment of malnutrition and in choosing the nutritional intervention for the patient—after assessing the nutritional risk using screening tools [65].

### 3.4. Quality Assessment Findings

Regarding the level of evidence based on the Oxford Centre for Evidence-Based Medicine, it is observed that more than half of the article sample was classified as high-level evidence based solely on the study design, as the majority of the studies were RCTs (*n* = 5; 62.5%) [57,58,59,60,61] classified as evidence level 1B. These studies were followed by quasi-experimental designs (*n* = 3; 37.5%) [62,63,64], classified at level 2B. We have identified six studies with 1B and two studies with 2B level of reference according to Oxford Centre for Evidence-based Medicine levels of evidence.

The assessment of the risk of bias was presented according to the study design and is represented in Figure 2. The majority of RCTs (*n* = 4; 80%) [57,59,60,61] were classified as low overall risk of bias, meaning that all domains were classified as “low risk of bias”. Only one RCT fell into the high overall risk of bias category [58] due to having “some concerns” in three domains (D1, D2, and D4). Figure 2A,B shows the assessment of bias risk for RCTs. It should be noted that bias due to missing outcome data or selective outcome reporting was not found in any of the RCTs (*n* = 5; 100%), so for both domains (D3 and D5), all RCTs were classified as low risk of bias according to RoB-2.

The analysis of risk of bias in nonrandomized intervention studies using the ROBINS-I tool (Table 2) indicated that all studies (*n* = 3; 100%) [62,63,64] presented a moderate risk of bias. The studies were classified as such mainly due to bias in participant selection as well as outcome measurement bias. In all quasi-experimental studies, the domains “bias in the classification of interventions”, “bias due to deviations from intended interventions”, and “bias in the selective reporting of outcomes” were well-reported and classified as low risk of bias.

## 4. Discussion

### 4.1. Omega-3 Supplementation

A growing body of studies has addressed the protective impact of omega-3 polyunsaturated fatty acids on patients with cancer, positively modulating the nutritional status in different types of malignant neoplasms, such as gastrointestinal [29], head and neck [28], and lung cancers [31]. The beneficial effects of omega-3 polyunsaturated fatty acid consumption are likely related to several mechanisms of action, including the following: (1) production of lipid mediators with pro-resolution properties such as resolvins and protectins, which are important modulators of inflammation [66]; (2) activation of free fatty acid receptors, specifically Free Fatty Acid Receptor 1 (FFA1) and Free Fatty Acid Receptor 4 (FFA4), leading to a decrease in cachexia-related outcomes [67]; (3) inhibition of nuclear factor kappa B (NF-κB) and reduction in the production of cytokines such as IL-6 and IL-8 [26].

These findings are consistent with the recent publication of the clinical nutrition guidelines for cancer by the European Society for Clinical Nutrition (ESPEN) in 2021, which recommend the use of long-chain omega-3 polyunsaturated fatty acid supplements (EPA and DHA) or fish oil in patients with advanced cancer undergoing chemotherapy and at risk of weight loss or malnutrition [22]. The proposed recommendations by some authors range from 2 to 4 g per day of omega-3 fatty acids (in a 2:1 ratio of EPA and DHA) [19,68], with a suggestion of 2 g/day of EPA for individuals with advanced cancer receiving chemotherapy and at risk of weight loss [23].

We conducted this systematic review based on eight clinical trials, including five RCTs and three quasi-experiments. Upon evaluating the clinical trials included in this review, we observed different forms of omega-3 supplementation administration, regarding the dosage, form, as well as timing of the administered omega-3 interventions. The offered doses of omega-3 varied from 0.3 g to 6 g per day of EPA, with three studies also supplementing DHA at doses ranging from 0.92 g to 0.96 g per day. In terms of the form of omega-3 intervention, two studies used capsules, while six studies provided a liquid supplement. The duration of supplementation ranged from 1 to 12 weeks in patients with resectable or unresectable pancreatic cancer. Despite the heterogeneity observed, none of the studies reported a worsening outcome for the patient groups receiving the supplement.

A recent systematic review of randomized clinical trials [31] investigating the efficacy of omega-3 in patients with lung cancer undergoing radiotherapy and chemotherapy also reported heterogeneity in the results. This review identified seven articles in which the variation in omega-3 dosage ranged from 0.51 g to 2.2 g per day of EPA. Among these studies, four also supplemented with DHA at doses ranging from 0.2 g to 0.92 g per day. The duration of supplementation varied from 5 to 12 weeks and concluded that omega-3 can improve the nutritional status as well as regulating inflammation indicators in these patients.

Regarding nutritional supplementation, Werner et al. [59] utilized omega-3 capsules containing 300 mg of DHA and EPA, derived from marine phospholipids (MPL) or fish oil (FO), for a duration of 6 weeks in patients undergoing treatment for pancreatic cancer. On the other hand, Wigmore et al. [64] employed capsules containing 500 mg of EPA (95% pure). The administration period started at 1 g/day in the first week, with a weekly increment of 1 g of EPA until reaching a total of 6 g/day for 12 weeks. The patients in this study were also undergoing treatment for their condition. Both studies reported on adherence to the nutritional supplementation via capsules. The group that took MPL capsules demonstrated higher adherence compared to the FO group. However, patients who supplemented with 95% pure EPA reported that the capsules were unpleasant to consume due to occasional leakage of their contents, resulting in an unpleasant chemical taste.

Administering a high dose of EPA to cancer patients in the form of capsules does not appear to be highly effective [69]. These findings were demonstrated by Fearon et al. [69] in a study that aimed to examine the effects of two doses of 95% pure EPA (2 g per day and 4 g per day) compared to placebo over a period of 8 weeks in the process of cachexia, in a double-blind randomized clinical trial involving 518 patients with gastrointestinal and lung cancer. The trial provides an explanation for the apparent lack of efficacy of 4 g of EPA/day, attributing it to the possible unobserved adverse effects, resulting in patients taking fewer capsules compared to those on the 2 g dose [69]. It should be highlighted that the identification of effective nutritional counseling strategies helps the nutritionist choose the most effective option for dietary treatment [70].

The most commonly used oral liquid nutritional supplementation was a dietary supplement enriched with protein, energy, and omega-3 fatty acids [60,61,62,63]. The interventions aimed to provide approximately 610–698 kcal, 32–36 g of protein, and 1.6–2.2 g of EPA per day, with durations ranging from 3 to 8 weeks. In addition to EPA, the two studies by Barber et al. [62,63] also included an approximate addition of 0.96 g of DHA. Patients tolerated the supplement well in the studies conducted by Fearon et al. [61] and Barber et al. [63].

Another systematic review by De Van Der Schueren et al. [71] identified 11 RCTs, including five trials conducted with oral nutritional supplements high in protein, high in energy, and enriched with omega-3 fatty acids for cancer patients undergoing antineoplastic treatment. This study yielded similar results to ours, both in terms of the intervention and adherence. The interventions aimed to provide approximately 590–600 kcal, 32–33 g of protein, and 2–2.2 g of EPA per day, with durations ranging from 5 to 12 weeks. Good adherence was reported in 3 RCTs, with two trials reporting adherence rates between 70% and 80%, and one trial reporting that 80% of participants consumed the prescribed dose [71].

### 4.2. Nutritional Status

Cancer patients often experience a reduction in their usual food intake due to anorexia, which can occur either as a primary symptom or as a result of secondary causes [23]. Additionally, approximately 71% of patients with pancreatic cancer have cachexia at the time of diagnosis [72]. Cachexia is recognized as a multifactorial syndrome defined by pathological weight loss caused by ongoing loss of lean muscle mass, with or without loss of adipose tissue, as a result of reduced food intake, anorexia, abnormal metabolism, or a combination of them [20].

The decreased and/or insufficient intake of the necessary nutrients to meet metabolic and physiological demands can lead patients to a pathological state of malnutrition, characterized by the intense catabolism of muscle and adipose tissues and continuous involuntary weight loss [73,74,75,76,77,78].

Additionally, the side effects of treatments such as chemotherapy, radiation therapy, hormone therapy, and immunotherapy also contribute to malnutrition [27]. The identification of malnutrition or the risk of malnutrition is indicative for assessing the nutritional status of the patient, allowing for the detection of possible nutritional disorders and enabling early intervention [24].

Among the studies that investigated the outcome of weight gain/maintenance, six demonstrated positive results [59,60,61,62,63,64]. Of these, four studies used a dietary supplement with protein, energy, and enriched with omega-3 fatty acids, which may partially explain the weight gain [60,61,62,63]. In the RCT by Barber et al. [62], which included 0.92 g of DHA in addition to EPA, a significant difference in the pattern of weight loss was found between the two groups (*p* = 0.0001), with the exposure group gaining an average of 1 kg while the control group lost an average of 2.8 kg. The study by Barber et al. [63], which also used an EPA supplement with 0.96 g of DHA, showed a mean weight gain of 1.0 kg (interquartile range 0.1 to +2.0, *p* = 0.024) compared to baseline and significant drop in IL-6 production. Clinical evidence demonstrating the isolated effects of DHA on the nutritional status of cancer patients is still scarce [68].

It is important to highlight that, among the four studies that gave positive results for weight gain, only Barber et al. [63] investigated IL-6 and, after the intervention, participants had a significant drop in the concentration of this substance in their blood. These positive findings may reflect the ability of omega-3 to modulate circulating inflammatory biomarkers such as IL-1, IL-6 and TNF-α, emphasizing its inhibitory effect on inflammatory parameters related to muscle atrophy and lipolysis [79]. Additionally, omega-3 is associated with the production of pro-resolving mediators such as resolvins and protectins, which decrease leukocyte infiltration, leading to the cessation of the inflammatory process [80]. This generates a positive balance between muscle anabolism and catabolism, thereby favoring weight gain/maintenance [68].

Findings from the systematic review of RCTs whose patients received omega-3 in the adjuvant treatment for lung cancer, have pointed out similar results by demonstrating that six out of seven studies reported changes in body weight, and the heterogeneity test showed *I*^2^ = 87.3%, *p* < 0.001 [31]. The assessment of the nutritional status in patients with pancreatic cancer can be performed using various, but low-cost and easy methods including body weight and lean body mass [24]. Low muscle mass has been associated with an increased incidence of toxicity, reduced effectiveness of chemotherapy in patients [24], worsened Karnofsky Performance Score [69], and negative impact on the quality of life of cancer patients [24]. However, the present systematic review identified only one study that reported a significant difference in patients who consumed 50% of the EPA-enriched supplement, showing significantly higher proportions of skeletal muscle mass compared to patients in the control group (*p* = 0.042) [58].

The transferrin investigation was conducted by both Barber et al. [62] and Ashida et al. [57]; however, only Barber et al. [62] showed positive results. The researchers concluded that acute phase proteins tend to progress in untreated patients but can be stabilized through the intake of a supplement enriched with fish oil, which will result in a reduction in atrophy in these patients [62]. The two studies that utilized capsules to supplement omega-3 showed similar results, with a significant stabilization of weight [59,64]. However, one study examined weight loss after 6 weeks of treatment in both groups compared to weight loss at baseline (GI: *p* = 0.001, GC: *p* = 0.003) [59]. The other study was a quasi-experiment lasting up to 12 weeks, and patients experienced a median weight gain of 0.5 kg compared to the rate of weight loss at the beginning of the study (*p* = 0.0009) [64].

These weight gain findings support the results of a randomized, triple-blind clinical trial that investigated the supplementation of omega-3 through capsules. The trial involved 40 women diagnosed with cervical cancer and showed a preservation of patients’ nutritional status and skeletal muscle quality. These results suggest that omega-3 supplementation can normalize some metabolic changes associated with cancer, which normally prevent patients from maintaining their body weight during treatment. [80,81] When evaluating the outcomes, only the study by Ashida et al. [57] did not show an improvement in the nutritional status after 7 days of omega-3 intake, as it did not demonstrate a significant difference in postoperative hypercytokinemia rates (*p* = 0.68) between the groups. Furthermore, it was the sole study with a 7-day preoperative intervention period. Despite not observing a significant impact from the supplementation, Ashida et al. [57] reported mean IL-6 values of 174.0 pg/mL and 148.2 pg/mL in the treatment and control groups, respectively, on the first postoperative day. Previous studies have shown that these values varied from 162 to 431 pg/mL [82,83].

### 4.3. Study Strengths and Limitations

This study has notable strengths. Firstly, it employed a systematic approach with high adherence to the PRISMA 2020 checklist, ensuring methodological rigor. The study demonstrated high transparency, particularly in the advanced search strategy. Additionally, the study protocol was previously published in a leading peer-reviewed journal, further enhancing its methodological robustness. Risk of bias assessment was conducted using current and revised tools from the Cochrane Collaboration [35]. However, due to study heterogeneity, it was not possible to conduct a meta-analysis of the data. The moderate quality of quasi-experimental studies is also a limiting factor, although 80% of the included RCTs exhibited low risk of bias and were well-reported. In addition, it is important to point out that, among the included studies, four used dietary supplements containing protein, energy, and enriched with omega-3 fatty acids; therefore, the data reported here require some caution. However, it is also important to highlight that all study participants received omega-3 supplementation, which is in accordance with the PICOS strategy of this systematic review. Nevertheless, this review provides valuable new insights to the literature by summarizing the available evidence specifically on this supplement for patients with pancreatic neoplasms.

## 5. Conclusions

In conclusion, the presented evidence indicates that omega-3 supplementation in patients with pancreatic cancer is safe, well-tolerated, and beneficial. It has been shown to stabilize or increase body weight, as well as reduce levels of inflammatory biomarkers. However, well-designed RCTs with low risk of bias are still needed to confirm these findings and establish an effective dosage of this supplementation for patients with pancreatic cancer. Furthermore, additional research is warranted to investigate the impact on lean mass gain and biomarkers of oxidative stress.

## Figures and Tables

**Figure 1 nutrients-16-04036-f001:**
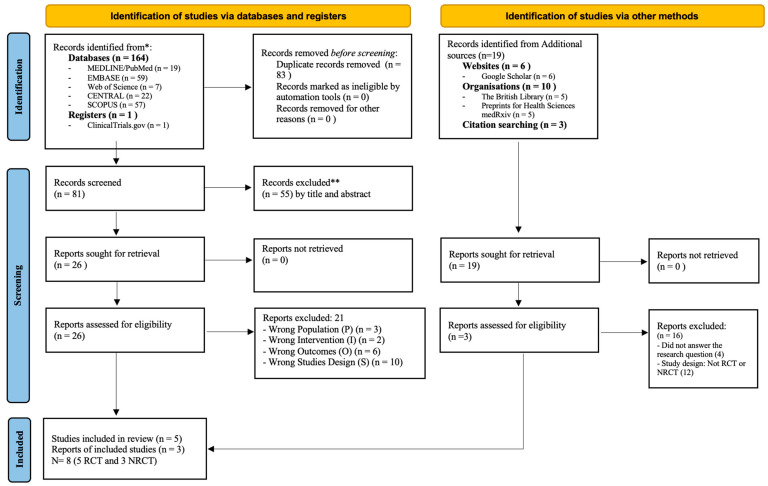
PRISMA Flowchart (* Consider, if feasible to do so, reporting the number of records identified from each database or register searched (rather than the total number across all databases/registers). ** If automation tools were used, indicate how many records were excluded by a human and how many were excluded by automation tools.).

**Figure 2 nutrients-16-04036-f002:**
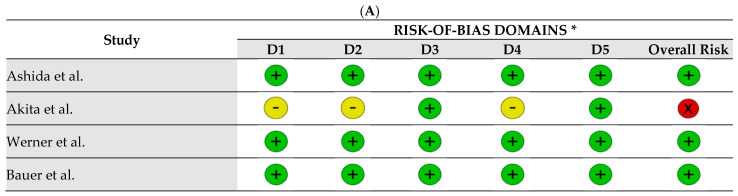
Summary of risk of bias judgements of RCT included according to RoB 2. (**A**) Internal validity and risk of bias assessment based on the RoB 2 [57,58,59,60,61]. (**B**) Percentage of risk of bias among RCT by domains of the RoB 2. (+) = low risk of bias; (-) = some concerns; (×) = high risk of bias. * Domains: D1: Bias due to randomization; D2: bias due to deviation from intended interventions; D3: bias due to missing data; D4: bias due to outcome measurement; D5: bias due to selections of reported results.

**Table 1 nutrients-16-04036-t001:** Characterization of the studies included in the systematic review.

Citation	Study Design	Aim	Sample	Type of Nutritional Supplement	Protocol	Follow-Up	Outcomes	Main Results	Conclusion
Ashida et al. [57] Japan	RCT	To investigate whether preoperative enteral diets enriched with EPA supplementation can reduce the incidence of hypercytokinemia	EG: (11), mean age = 64 (SD = 11) GC:(9), mean age = 69 (SD = 6)	Liquid	EG: supplementation (600 kcal/day) containing 2.0 g/d of EPA + 1200 kcal of regular food GC: standard isocaloric, isonitrogenous diet (600 kcal/d) without EPA + 1200 kcal of regular food	Supplementation for 7 days preoperatively	Primary: postoperative serum IL-6 concentration Secondary: postoperative nutritional status (serum albumin, prealbumin, transferrin, and EPA/AA ratio) and incidence of operative infectious complications	No statistically significant differences were identified in serious levels of IL-6 (*p* = 0.68), serum albumin (*p* = 0.56), prealbumin, transferrin (*p* = 0.65) and EPA/AA ratio after intervention	EPA supplementation had no marked impact on postoperative hypercytokinemia and nutritional status
Akita et al. [58] Japan	RCT	To clarify the usefulness of EPA-enriched supplementation during neoadjuvant chemoradiotherapy	EG: (31), mean age = 67.8 (SD = 10.7) GC: (31), mean age = 66.4 (SD = 9.8)	Liquid	EG: 2 vials (440 mL) (560 kcal) of EPA (Prosure^®^; Abbott Japan) per day + normal diet + 3 nutritional consultations (before, 3 weeks, and after radiotherapy). CG: Normal diet + 3 nutritional consultations (before, 3 weeks, and after radiotherapy).	During neoadjuvant chemoradiotherapy (about 5 weeks)	Primary: post/pre ratio of skeletal muscle mass and PMA cm^2^ Secondary: nutritional parameters (serum prealbumin, serum albumin, BMI, and lymphocyte count	Only 45.2% of the EG patients consumed more than 50% of the supplement with EPA. No significant differences in nutritional parameters and skeletal muscle mass post-pre ratio were observed. However, patients who consumed 50% of the supplement had significantly better values than the CG. The PMA post/pre ratio was significantly higher in EG.	EPA supplementation can potentially improve the nutritional status of patients with pancreatic cancer under neoadjuvant chemoradiotherapy
Werner et al. [59] Germany	RCT	To compare low-dose MPL and FO supplementation with the same amount of omega-3 in stabilizing weight.	EG FO: (18), mean age = 70.3 (SD = 8.24) EG MPL: (15), mean age = 71.3 (SD = 7.51)	Capsule	EG FO:500 mg capsule 3× daily: 60% FO, 40% MCT (6.9 g/100 g EPA and 13.6 g/100 g DHA) EG MPL: 500 mg capsule 3× daily: 35% omega-3 fatty acids phospholipids (mainly phosphatidylcholine) + 65% lipids neutral (8.5 g/100 g EPA and 12.3 g/100 g DHA); The final dose of omega-3 was 300 mg/day in both groups	During chemotherapy, radiotherapy, palliative care for 6 weeks	Primary: change in weight Secondary: nutritional status and quality of life	Weight stabilization EG FO (*p* = 0.001) and EG MPL (*p* = 0.003). Significant increase in EPA in plasma triglycerides of EG FO (*p* = 0.001) and EG MPL (*p* = 0.01)	The administration of omega-3 as FO or MPL is highly accepted, resulting in weight stabilization, and reflected in the increase in omega-3 in plasma lipids. However, MPL was better tolerated and accepted in the study group. In both groups, there were no significant changes in quality of life after 6 weeks.
Bauer et al. [60] (Australia)	RCT	To evaluate the effect of adherence to the nutritional prescription of an oral nutritional supplement with omega-3 and dense in protein and energy	EG: (87), mean age = 66.87 (SD = 1.0) GC MPL: (98), mean age = 68.37 (SD = 1.1)	Liquid	EG: 2 cans/day of a nutritional supplement rich in protein and energy + omega-3 fatty acids (1.1 g EPA). GC: isocaloric and nitrogenated control supplement without omega-3 fatty acids. Consume 1.5 cans/day (465 kcal and 24 g of protein) for both groups.	8 weeks	Primary: body composition and food intake Secondary: quality of life	There was a significant difference in energy and protein intake between patients in the EG compared to the CG (*p* < 0.05). The EG had an increase in weight of 0.5 kg compared to the CG, which decreased by 0.7 kg.	Adherence to the prescription of a protein- and energy-dense oral omega-3 supplement improved outcomes related to the nutritional status of patients with pancreatic cancer
Fearon et al. [61] Multicentric (UK, Netherlands, Canada, Italy, Belgium, and Australia)	RCT	To compare an omega-3-enriched protein and energy-dense supplement with an isocaloric, isonitrogenous supplement	EG: (95) mean age = 67 (SD = 1.0) CG: (105) mean age = 67 (SD = 1.0)	Liquid	EG:2 cans/day of omega-3-enriched protein and energy-dense supplement (480 mL, 620 kcal, 32 g protein, 2.2 g EPA) CG: 2 cans/day of the supplement (480 mL, 620 kcal, 32 g protein) without EPA	8 weeks	Primary: Weight, lean body mass, and food intake Secondary: quality of life	Compared with baseline loss rates, weight, and lean body mass loss were significantly attenuated in both study groups at four and eight weeks (*p* < 0.001 for all group comparisons).	At the average dose taken, enrichment with EPA did not provide a therapeutic advantage, and both supplements were equally effective in halting weight loss. The post hoc dose–response analysis suggests that, if taken in sufficient quantity, only the EPA-rich supplement results in weight gain and lean body mass. Weight gain was only associated with improved quality of life (*p* < 0.01) in EG.
Barber et al. [63] Glasgow, UK	NRCT	To examine the effect of a fish oil-enriched nutritional supplement on various mediators believed to play a role in cancer cachexia.	EG: (20) Median age = 62 (min/max: 51–75)	Liquid	EG: 2 cans/day supplement enriched with fish oil (2.2 g EPA and 0.96 g DHA)	3 weeks	Primary: weight Secondary: serum concentrations of interleukin IL-6, TNF-a, cortisol, insulin, and leptin	Significant drop in IL-6 production (*p* = 0.015), increase in the serum insulin concentration (*p* = 0.0064), decrease in the cortisol/insulin ratio (*p* = 0.0084), and these changes occurred in association with weight gain (median 1 kg, *p*= 0.024)	Several mediators of catabolism in cachexia were modulated by the administration of a nutritional supplement enriched with fish oil in patients with pancreatic cancer, which may explain the reversal of loss of weight in these patients
Wigmore et al. [64] Scotland, UK	NRCT	To evaluate the acceptability and effects of oral supplementation with high-purity EPA in patients with advanced pancreatic cancer on weight loss	EG: (26), Median age = 56 (min/max: 39–75)	Capsule	EG: capsule containing 500 mg of EPA	EPA at 1 g/day in the 1st week, 2 g/day in the 2nd week, 4 g/day in the 3rd week, and 6 g/day after that	Weight and body composition	Supplementation was well tolerated. After supplementation, the weight remained stable. After 4 weeks of EPA supplementation, patients had a median weight gain of 0.5 kg (*p* = 0.0009 vs. baseline weight loss rate), and this weight stabilization persisted through the study period. 12-week study.	EPA is well tolerated, may stabilize weight in patients with cachectic pancreatic cancer, and should be tested as an anti-cachectic agent in RCT
Barber et al. [62] Scotland, UK	NRCT	To determine the effects of administering a nutritional supplement containing EPA-rich fish oil on acute-phase protein response levels	EG: (18) Median age = 64 (min/max: 56–66) CG: (18) Median age = 60 (min/max: 54–70) Healthy CG: (6) Median age = 54 (min/max: 50–56)	Liquid	EG: 2 cans/day supplement enriched with fish oil (2.18 g EPA and 0.92 g DHA) in a volume of 480 mL CG: full support without EPA and DHA supplementation	3 weeks	Primary: Positive and negative acute phase protein Minor: weight	Increased transferrin in EG (*p* = 0.048). In the CG, there was a reduction in albumin, transferrin, and prealbumin(*p* = 0.012; *p* = 0.0048 and *p*= 0.038, respectively) and increase in the positive concentration of CRP (*p* = 00013). The EG gained an average of 1 kg, and the CG lost an average of 2.8 kg of body weight	Acute-phase CRP can be stabilized by administering a nutritional supplement enriched with fish oil. This may have implications for reducing atrophy in these patients.

Abbreviations: RCT: randomized controlled trial; NRCT: nonrandomized controlled trial; EG: experimental group; CG: control group; SD: standard deviation; IL-6: interleukin-6; EPA: eicosapentaenoic acid; DHA: docosahexaenoic acid; AA: arachidonic acid; BMI: body mass index; PMA: cm^2^ area of the greater bilateral psoas muscle; MPL: marine phospholipids; FO: fish oil; MCT: medium-chain triglycerides; CRP: C-reactive protein; TNF-a: tumor necrosis factor-alpha.

**Table 2 nutrients-16-04036-t002:** Evaluation by consensus of ROBINS-I between two reviewers per bias domain.

	* Domains ROBINS-I							Overall Judgment ROBINS-I
Study	Confounding Bias	Participant Selection Bias	Classification of Intervention Bias	Bias Due to Intervention Deviations	Incomplete Data Bias	Outcome Measurement Bias	Selective Outcome Reporting Bias	
Barber et al. [63]	Low	Moderate	Low	Low	Moderate	Moderate	Low	Moderate
Wigmore et al. [64]	Low	Moderate	Low	Low	Low	Moderate	Low	Moderate
Barber et al. [62]	Moderate	Moderate	Low	Low	Low	Moderate	Low	Moderate

Acronyms: * ROBINS-I, risk of bias in non-randomized studies [54].

## Data Availability

Data are available upon reasonable request due to privacy.

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
