# Peer review of "Omega-3 Supplementation and Nutritional Status in Patients with Pancreatic Neoplasms: A Systematic Review"

_nutrients, 2024, doi:10.3390/nu16234036_

Round 1

Reviewer 1 Report

Comments and Suggestions for Authors

To the Authors,

The systematic review entitled “Omega-3 Supplementation and Nutritional Status in Patients 2 with Pancreatic Neoplasms: A Systematic Review” poses the question of the existing evidence the benefits of omega-3 supplements on the nutritional status of patients with pancreatic cancer following the recommendations of European Society for Medical Oncology (ESMO) – “body composition, BMI, dietary intake, C-reactive protein, albumin, systemic inflammation, subjective global assessment (SGA), and patient-generated subjective global assessment (PG-SGA)”. The authors identified ‘the lack of summarized evidence’ as the main gap of knowledge in this area of research. By answering this question, the study would provide evidence in using these supplements in clinical settings. The introduction is clearly formulated and presents sufficient background to motivate the analysis.

The Methods section addresses the points included in PRISMA 2020  - Eligibility criteria , Information sources , Search strategy, Selection process, Data collection process, Study risk of bias assessment, Synthesis methods,  Reporting bias assessment. My main comment addresses the question of how the authors tested for Inter-rater Reliability or other methods of Agreement.

In PRISMA figure – what is CENTRAL?.

In the text, the authors did not mention using the automation tool for the selection of the studies. In my opinion this should be mentioned in text and excluded from the PRISMA figure.

The authors identified 6 studies with 1B and two studies with 2B level of reference according to Oxford Centre for Evidence-based Medicine levels of evidence and mentioned this in the Quality Assessment Findings. In my opinion, these findings could be mentioned in the text and removed from the table to make it more easy to read.

Line 174-176 “The study also reported that many patients were unable to consume the recommended amount of the EPA-enriched supplement due to its taste” -please provide more precise data (percent).

“Figure 3. Evaluation by consensus of ROBINS-I between two reviewers per bias domain.” – It has the format of a table.

The Results and Discussion sections are well organized and provide sufficient data related to the effects of omega 3 acids on the nutritional aspects included in the European Society for Medical Oncology (ESMO) and mentioned in the introduction section.

Minor revisions

Line 93 –  there is a typo - “Emtree terms”

PRISMA figure is not very clear.

Provide reference for *Oxford Centre for Evidence-based Medicine levels of evidence.

 The reference number in the text should be modified according to the journal recommendation.

The reference format should be in the MDPI format. 

Author Response

November 10th 2024

Dear Editor/Reviewer of Nutrients

We would like to express our gratitude for the valuable comments and suggestions provided to improve our manuscript. We carefully reviewed each recommendation and have implemented all the suggested changes.

Reviewer 1 :

Comments and Suggestions for Authors

To the Authors,

The systematic review entitled “Omega-3 Supplementation and Nutritional Status in Patients 2 with Pancreatic Neoplasms: A Systematic Review” poses the question of the existing evidence the benefits of omega-3 supplements on the nutritional status of patients with pancreatic cancer following the recommendations of European Society for Medical Oncology (ESMO) – “body composition, BMI, dietary intake, C-reactive protein, albumin, systemic inflammation, subjective global assessment (SGA), and patient-generated subjective global assessment (PG-SGA)”. The authors identified ‘the lack of summarized evidence’ as the main gap of knowledge in this area of research. By answering this question, the study would provide evidence in using these supplements in clinical settings. The introduction is clearly formulated and presents sufficient background to motivate the analysis.

The Methods section addresses the points included in PRISMA 2020  - Eligibility criteria , Information sources , Search strategy, Selection process, Data collection process, Study risk of bias assessment, Synthesis methods,  Reporting bias assessment.

Response: Thank you so much for your positive feedback!

My main comment addresses the question of how the authors tested for Inter-rater Reliability or other methods of Agreement.

Response: The process of study selection, data extraction and risk of bias assessment was carried out by two reviewers independently as recommended by the Cochrane Handbook. In addition, for the article selection process, blinded assessment was organized and managed in the Rayaan App. After the completion of the individual and independent assessment by the authors (LBCP and LCLJ), blinding was revealed and divergences were identified. The decision to include the study or not was made after discussion and extensive analysis of the article, as well as by a third reviewer who was an expert in systematic review studies.

In PRISMA figure – what is CENTRAL?

Response: Cochrane Central Register of Controlled Trials (CENTRAL) Cochrane. I have added this information on Methods section. Thanks!

In the text, the authors did not mention using the automation tool for the selection of the studies. In my opinion this should be mentioned in text and excluded from the PRISMA figure.

Response: We have mentioned this on Methods section. “During this stage of article inclusion and exclusion, the Rayyan™app43 was used as a tool to assist in the eligibility phase”

The authors identified 6 studies with 1B and two studies with 2B level of reference according to Oxford Centre for Evidence-based Medicine levels of evidence and mentioned this in the Quality Assessment Findings. In my opinion, these findings could be mentioned in the text and removed from the table to make it more easy to read.

Response: OK. Done! Thank you for this suggestion.

Line 174-176 “The study also reported that many patients were unable to consume the recommended amount of the EPA-enriched supplement due to its taste” -please provide more precise data (percent).

Response: OK. It was 54.83%

“Figure 3. Evaluation by consensus of ROBINS-I between two reviewers per bias domain.” – It has the format of a table.

Response: OK. We have changed to Table 2 instead of Figure 3.

The Results and Discussion sections are well organized and provide sufficient data related to the effects of omega 3 acids on the nutritional aspects included in the European Society for Medical Oncology (ESMO) and mentioned in the introduction section.

Response: Thank you so much for your positive feedback!

Minor revisions

Line 93 –  there is a typo - “Emtree terms”

Response: “Emtree terms” está correto, e é o descriptor controlado da base embase (https://www.elsevier.com/products/embase/emtree)

PRISMA figure is not very clear.

Response: This PRISMA Figure is standardized by the PRISMA statement 2020 for this type of review that has databases, records and Other sources.

Provide reference for *Oxford Centre for Evidence-based Medicine levels of evidence.

Response: OK. Done! Oxford Centre for Evidence-Based Medicine. Levels of evidence work- ing group. “The Oxford 2011 Levels of Evidence”. Oxford centre for evidence-based medicine. Oxford, UK: Centre for Evidence-Based Medicine (CEBM); 2011.

 The reference number in the text should be modified according to the journal recommendation.

Response: OK. Done!

The reference format should be in the MDPI format. 

Response: OK. Done!

Reviewer 2 Report

Comments and Suggestions for Authors

Authors gave us a very important study and systematic review on Omega-3 supplementation and nutritional status in patients with pancreatic neoplasms. They did a systematic review and synthesize and evaluate the evidence regarding the effects of omega-3 supplementation on the nutritional status of pancreatic cancer patients. The manuscript is logic and study design is also novelty. It is meaningful in this study field. However, before getting to the publication standards, some critical issues should be clarified as follows:

1.      In the part of introduction, the omega-3 study progress should be introduced as the supplementation in pateints.

2.      In addition to the data in the database, is there any data of first-line hospitals as support to the study?

3.      The reason of “The evidence level was classified using the Oxford Centre for Evidence-Based Medicine” should be explained, to myself, I don’t know this methodological assessment.

4.      In part of 3.2, why the age is only from 58-78 years?

5.      In the part of discussion, many mechanism of omega-3 supplement to patients such as cytokines, receptor were discussed, but there is no relatively systematic review data in the part of results.

6.      And more references should be cited in the part of discussion.

7.      Authors should expanded the part of conclusion.

Author Response

Reviewer 2 :

Comments and Suggestions for Authors

Authors gave us a very important study and systematic review on Omega-3 supplementation and nutritional status in patients with pancreatic neoplasms. They did a systematic review and synthesize and evaluate the evidence regarding the effects of omega-3 supplementation on the nutritional status of pancreatic cancer patients. The manuscript is logic and study design is also novelty. It is meaningful in this study field.

Response: Thank you so much for your positive feedback!

However, before getting to the publication standards, some critical issues should be clarified as follows:

  1. In the part of introduction, the omega-3 study progress should be introduced as the supplementation in patients.

Response: OK. Done!

  1. In addition to the data in the database, is there any data of first-line hospitals as support to the study?

Response: No.

  1. The reason of “The evidence level was classified using the Oxford Centre for Evidence-Based Medicine” should be explained, to myself, I don’t know this methodological assessment.

Response: The Oxford Centre for Evidence-Based Medicine” was not used for methodological assessment and in fact should not be used for this. We only used Oxford to classify the level of evidence based solely on the study design (this part is optional in review studies). The methodological assessment was done by ROB-2 and ROBINS-I, both of which assess the internal validity and risk of bias of RCTs and NRCTs, respectively.

  1. In part of 3.2, why the age is only from 58-78 years? 

Reponse: Because this is the age range that the 8 studies included in this systematic review addressed in their samples.

  1. In the part of discussion, many mechanism of omega-3 supplement to patients such as cytokines, receptor were discussed, but there is no relatively systematic review data in the part of results.

Response: The data regarding biomarkers are explicit in Table 1 in the “Main results” section and were therefore discussed.

  1. And more references should be cited in the part of discussion.

Response: OK. Done. We have added more 5 references

  1. Authors should expanded the part of conclusion.

Response: We have already covered everything that can be concluded in the study based on the research question.

Thank you  for your dedication and the detailed suggestions, which were essential in enhancing the quality of our work.